# Liver-Gut-Interaction: Role of Microbiome Transplantation in the Future Treatment of Metabolic Disease

**DOI:** 10.3390/jpm13020220

**Published:** 2023-01-27

**Authors:** Vanessa Stadlbauer

**Affiliations:** 1Department of Gastroenterology and Hepatology, Medical University of Graz, 8036 Graz, Austria; vanessa.stadlbauer@medunigraz.at; 2Center for Biomarker Research in Medicine (CBmed), 8010 Graz, Austria

**Keywords:** microbiome, obesity, metabolic syndrome, fecal microbiota transplantation

## Abstract

The association between shifts in gut microbiome composition and metabolic disorders is a well-recognized phenomenon. Clinical studies and experimental data suggest a causal relationship, making the gut microbiome an attractive therapeutic goal. Fecal microbiome transplantation (FMT) is a method to alter a person’s microbiome composition. Although this method allowed for the establishment of proof of concept for using microbiome modulation to treat metabolic disorders, the method is not yet ready for broad application. It is a resource-intensive method that also carries some procedural risks and whose effects are not always reproducible. This review summarizes the current knowledge on FMT to treat metabolic diseases and gives an outlook on open research questions. Further research is undoubtedly required to find applications that are less resource-intensive, such as oral encapsulated formulations, and have strong and predictable results. Furthermore, a clear commitment from all stakeholders is necessary to move forward in the direction of developing live microbial agents, next-generation probiotics, and targeted dietary interventions.

## 1. Introduction

Obesity and the metabolic syndrome lead to a considerable disease burden due to complications, such as cardiovascular disorders, cancer, dementia, and fertility disorders, and are also associated with socio-economic disadvantages [1]. Obesity and its relation to shifts in the microbiome composition were among the first described associations when culture-independent techniques to study the complex ecosystem of the gut microbiome were developed about 20 years ago [2]. Starting from a description of shifts in microbiome composition at the phylum level, a large body of literature has evolved that describes the bidirectional interaction between the gut microbiome and human metabolism [3]. Human association studies clearly demonstrated an association between the gut microbiome composition and obesity. The causal relationship was first suggested by the finding that the phylum Bacteroides was reduced in obesity and that weight loss through diet led to an increase [4,5]. Data from in vitro systems and animal models suggest that diet or drugs (e.g., antibiotics) alter the microbiome, and this dysbiosis is mechanistically involved in disrupting molecular metabolism and signaling through bacterial metabolites (e.g., bile acids), which impacts energy intake and leads to metabolic disorders. For example, a high-fat and high-glucose diet lead to increased gut permeability, translocation of bacterial products, a low grade inflammatory response, and insulin resistance, indicating that the gut-liver axis is involved in the metabolic sequelae of the western diet [6,7]. Certain metabolites, such as bile acids, that are bidirectionally interacting with the microbiome have been identified as playing a mechanistic role in the development of metabolic syndrome via the gut microbiome. In obesity, altering bile acid composition via reduction of microbial diversity through an antibiotic leads to increased insulin resistance [8]. Furthermore, host genetics play a role since immune control of the microbiome maintains beneficial microbial populations that constrain lipid metabolism to prevent metabolic syndrome [9]. The strongest evidence for a causal relationship between the gut microbiome and obesity is derived from animal experiments in germ-free mice, which received fecal transplantation from human twins discordant in obesity. The mice transplanted with the microbiome of the obese twin developed obesity, while the mice who received the microbiome of the lean twin stayed lean [10]. As a result of this strong link between the gut microbiome and metabolic disorders, new therapeutic modalities targeting the altered commensal bacteria as a means of treating metabolic syndrome have gained a lot of interest.

## 2. Fecal Microbiota Transplantation (FMT) for Metabolic Syndrome

FMT, the transfer of fecal matter from one individual to another with the aim to improve/restore the composition of the gut microbiome and thereby treat a disease, has gained much attention in the scientific field and also in the general public. As a method that in principle dates back to the fourth century in China, its modern application took off in 2013. Until now, the only routine medical application was treatment of recurrent *Clostridioides difficile* infection. But also, many non-infectious diseases have been extensively studied, such as inflammatory bowel diseases, the microbiota-gut-liver axis, the microbiota-gut-brain axis, and oncologic and hematological diseases [11,12]. The first indication of efficacy for FMT in human metabolic diseases was published in 2012, where in a pilot study in the Netherlands, in nine male adults with metabolic syndrome who received FMT from a lean donor, insulin resistance was improved, whereas no changes were observed in nine controls who received autologous FMT [8]. A potential effect on body weight was first suspected by a “complication” of FMT for *Clostridioides difficile,* where a patient with normal weight received an FMT from an obese donor and gained weight after FMT [13]. While consecutive studies did not report such effects [14], the opposite strategy of increasing the body weight of cachectic cancer patients through FMT from obese donors failed [15]. Nevertheless, body mass index (BMI) in the normal range has been included in the selection criteria for stool donors after the suspicion that obesity could be actually “transferred” by FMT [16].

After these initial proof of concept trials, a number of clinical trials have been performed, where researchers aimed to improve metabolism and reduce weight by FMT in obesity, metabolic syndrome, non-alcoholic fatty liver disease and type 2 diabetes. However, especially in adequately powered, randomized, placebo-controlled trials, mixed results with regards to improvement in metabolic parameters were reported. While most studies demonstrate that FMT is able to change the microbiome composition, clear effects on clinically important endpoints, such as body weight or insulin resistance, are missing. Trials with clinical endpoints are summarized in Table 1. It is notable that the method of FMT may play a role in its effectiveness. Oral capsules, which would be the preferred route of administration for both safety and logistic reasons, have yet to show substantial improvements in metabolic parameters, except when administered together with low fermentable fibers [17,18,19,20]. From a mechanistic point of view, it is still not clear what the “effective agent” in FMT is. The notion that bacteria might be the “effective agent” has been challenged, since for the treatment of *Clostridioides difficile* infection, sterile filtrated fecal preparations were similar to or more effective than conventional fecal preparations containing living microorganisms [21]. This indicates that maybe not the living bacteria, but rather bacterial components, metabolites, or bacteriophages may mediate the effects of FMT. From a pilot study, it was suggested that bacteriophages in the recipient that are associated with metabolic syndrome are replaced by new viruses that were not originally present in either the healthy donors or control subjects at detectable abundances and that changes in the bacteriophage population are related to response to FMT [22]. Another study reported an influence of FMT on plasma metabolites related to lipid metabolism and DNA methylation status; however, a clear-cut pathophysiological explanation for a potential mechanism to influence glucose metabolism could not be identified [23]. A recent systematic review did not identify consistent changes in clinically relevant endpoints (such as insulin sensitivity) achieved in the recipient after FMT for metabolic diseases [24].

## 3. Going beyond FMT

Recent studies also explored how the selection of the donor could impact the effect of FMT. A vegan diet is associated with reduced trimethylamine-N-oxide (TMAO) production and therefore lower cardiovascular risk; however, the FMT from vegan donors did not decrease TMAO production in patients with metabolic syndrome, which indicates that despite evidence of compositional changes that resemble the microbiome of the donor, the functional capacity is not easily transferred [33]. FMT from bariatric surgery patients and obese patients with metabolic syndrome, revealed microbiome-driven modulation of brain dopamine and serotonin transporters [34]. And the use of autologous fecal transplants with fecal material obtained at the “weight nadir” of a successful diet was able to delay weight regain after the diet [25,26]. These studies indicate that the concept of FMT most likely needs to be augmented by adequate preparatory measures like diet or additional prebiotic “fertilizers” of the transplanted microbiome. Since studies also show considerable differences between different donors, it is essential to characterize donors and understand the interaction between the dysbiotic recipient microbiome and the donor microbiome [35]. The host microbiome composition determines the efficacy of engraftment of an FMT [36]. “Tuning” the host microbiome, e.g., by special dietary measures such as fiber supplementation, may improve functional engraftment of FMT [19,29].

## 4. Challenges in FMT for Metabolic Diseases

When transferring living microorganisms to a new host, adverse events have to be considered. It is surprising that some studies do not report safety data (see Table 1). A meta-analysis on the safety of FMT across different disease entities showed no significant differences in the incidence of adverse events between FMT and the control group. Adverse events can be related to the transplanted microbiome or to the route of administration. It seems that administration via oral capsules or endoscopically via the lower gastrointestinal tract is less prone to adverse events. Translocation and infection with transplanted bacteria can occur; the risk seems to be higher in patients with an altered intestinal barrier [37]. In the studies related to metabolic diseases, no bacteremia or sepsis events have been described so far.

From a practical point of view, however, it is unlikely that FMT, which requires highly skilled personnel and is resource intensive, will be applicable to treat the worldwide “obesity pandemic”. Although no international figures exist for the workforce requirements to perform large-scale FMT treatments, a shortage in skilled personnel can be extrapolated from studies related to cancer screening colonoscopies, where a severe shortage was noticed [38].

FMT clearly helps to understand the relationship between the gut microbiome and metabolic disorders and facilitates the notion that microbiome modulation can be an effective therapeutic strategy. The low predictability and the resource intensity of the intervention warrant further research towards a better translation or transformation into clinical practice. Further efforts are necessary to improve the timely and personalized diagnosis of the individual dysbiosis in obesity and augment and retain the effect of a diet by influencing the microbiome in a personalized but also “affordable” microbiome modulation strategy. A clear commitment to research from all stakeholders (politics, funding bodies, the health industry, researchers, and society) is necessary to move forward in the direction of developing live microbial agents, next-generation probiotics, and targeted dietary interventions. Table 2 summarizes the pros and cons for FMT in metabolic disorders.

## Figures and Tables

**Table 1 jpm-13-00220-t001:** Clinical studies on FMT to treat obesity and metabolic syndrome.

Study	Population	Number of Participants	FMT Mode	FMT Duration and Dose	Primary Endpoint	Result Metabolic	Result Microbiome	Adverse Events
Yu 2020 [20]	obesity and insulin resistance	24 adults	oral capsules	6 weeks, once per week versus placebo	insulin sensitivity	no change in insulin sensitivity	shift towards donor microbiome	no significant difference to placebo, no severe adverse events
Allegretti 2020 [17]	obesity	22 adults	oral capsules versus placebo	8 weeks, 2 doses	safety	no reduction in BMI	shift towards donor microbiome	no significant difference to placebo
Rinott 2021 [25,26]	obesity or dyslipidemia, randomized to healthy dietary guidelines, Mediterranean diet, and green-Mediterranean diet weight-loss groups	90 adults	autologous transplantation of microbiome collected under diet	100 capsules in 8 months versus placebo	weight regain	autologous FMT attenuated weight regain in combination with green Mediterranean diet	green Mediterranean diet caused change in microbiome composition	no treatment attributable adverse events
Mocanu 2021 [19]	obesity and metabolic syndrome	70 adults	oral capsules, + either high fermentable or low fermentable fibres versus placebo	6 weeks, 1 dose	insulin sensitivity	FMT + low fermentable fibers improved insulin sensitivity	FMT + low fermentable fibers increased diversity, shift towards donor microbiome	no treatment attributable adverse events
Leong 2020 [18,27]	obesity	87 adolescents	oral capsules	6 weeks, 1 dose versus placebo	BMI	no effect on BMI, reduction of abdominal obesity, resolution of metabolic syndrome at baseline	greater dissimilarity between baseline and post treatment in FMT group versus placebo, increase in diversity in female participants	no treatment attributable adverse events
Craven 2020 [28]	non-alcoholic fatty liver disease	21 adults	allogenic or autologous FMT in the distal duodenum via endoscopy	6 weeks, 1 dose	insulin resistance	no effect on insulin resistance, reduction of increased intestinal permeability	no changes	not reported
Ng 2022 [29]	type 2 diabetes	61 adult	allogenic FMT via nasogastric tube +, lifestyle intervention	24 weeks, 3 doses	donor microbiome engraftment	FMT + lifestyle intervention reduced total and low-density lipoprotein cholesterol and liver stiffness (secondary endpoints)	FMT + lifestyle intervention significantly better engraftment (primary endpoint)	no differences between groups, several cardiovascular events
Xue 2022 [30]	non-alcoholic fatty liver disease	75 adults	allogenic FMT, 3 doses, 1 via colonoscopy 3 via enema, versus oral probiotics	4 weeks, 3 doses	clinical efficacy and safety (not further specified)	no increase in liver fat content compared to probiotic group	differing responses in lean and obese patients	not reported
Ding 2022 [31]	type 2 diabetes	17 adults	allogenic FMT, transendoscopic jejunal tube, unblinded	12 weeks, 2 doses	insulin resistance	improvement in HbA1c, glucose, uric acid, increase in C-reactive protein	difference between responders and non-responders	no adverse events
Su 2022 [32]	type 2 diabetes	16 adults	allogenic FMT + formula diet versus formula diet alone, oral capsules	12 weeks, 3 doses	health status	Both interventions improved BMI, glucose metabolism and blood pressure	reduction of diversity in both groups, less in FMT group	no adverse events

FMT: fecal microbiome transplantation, BMI: body mass index.

**Table 2 jpm-13-00220-t002:** Pro and con arguments for/against FMT to treat metabolic diseases.

PRO	CON
Altered microbiomes are likely to contribute to the pathophysiology of metabolic diseases	Our conceptual, mechanistic and ecological understanding of FMTs is poorly developed
Experiments in animal models provide a rationale for FMTs because the clinical phenotype can be transplanted and reverted by FMT	Difficult to rationalize timing and dosing regimens, and it is also unclear whether dysbiosis can be corrected
Known dysbiosis in metabolic diseases provides a rationale for “complete” microbiome restoration through FMT	Safety concerns (both short- and long-term) regarding the exposure to infectious and non-infectious transmissible diseases
FMT has the potential to revert aspects of dysbiosis, engraft health promoting microbes and/or expose the host temporarily to beneficial microbes	Risk of conveyance of unintended characteristics of the donor microbiome that might predispose to chronic diseases
There is already one standard use case for FMT (refractory C. difficile infection)	It is currently unclear which specific measures are necessary to enable FMT to work effectively
Broad acceptance from patients	Application via colonoscopy is costly and bears a small but relevant procedural risk
	Other routes of application, e.g., encapsulation, do not yet provide satisfactory efficacy

## Data Availability

Not applicable.

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
