# Peer review of "Liver-Gut-Interaction: Role of Microbiome Transplantation in the Future Treatment of Metabolic Disease"

_jpm, 2023, doi:10.3390/jpm13020220_

Round 1

Reviewer 1 Report

The authors provided with their review an overview of current experiences with FMT to treat metabolic disease, whereby they identify possible mechanisms, knowledge gaps, requirements and challenges. As such it is a comprehensive snapshot of current status from which a number of subsequent efforts are described to overcome challenges to treat metabolic diseases with dietary and microbial interventions.

Author Response

I would like to thank the reviewer for his/her time to read the manuscript

Reviewer 2 Report

The review paper entitled: Liver-Gut-Interaction: role of microbiome transplantation in the future treatment of metabolic disease; performs a compilation of experiments on fecal microbiota transplantation aimed at adequate maintenance of body mass index, or change in insulin resistance, improvement of metabolic syndrome indicators.

In general, the author proposes a state of knowledge in which it is necessary to explore the ideal conditions of the donor and the transplant method so that it is efficient. His tables are clear about the inconsistency of the desired results, for which he clearly states that the fecal microbiota transplant procedure has not been optimized.

The review is well-written, simple, and, therefore, state-of-the-art work. However, I would only recommend improving the separation of columns in tables 1 and 2 since some texts are confused. Additionally, in line 69, where the initials BMI are entered, define their meaning, which should be Body Mass Index due to the context.

An application of fecal microbiota transplantation that is not touched on by the review but is of clinical importance is for treating intestinal lesions, in which authors such as Thomas J. Borody have made essential contributions since the 1980s. Therefore, it would be advisable for the review to touch on the subject to distinguish itself from several other contemporary studies that also do not touch on the subject.

Author Response

I would like to thank the reviewer for his/her time to review the manuscript.

I separate the columns in the tables

I spelt out BMI and added the abbreviation to the table

I added a sentence on other non-infectious indications fro FMT and added a recent review from Thomas J. Borody as suggested.